



# Geomagnetic pulsations in the Pc5/Pi3 frequency range and fluctuations of foF2 frequency

Nadezda Yagova[1], Alexander Kozlovsky[2], Evgeny Fedorov[1], and Olga Kozyreva[1]

[1]Schmidt Institute of Physics of the Earth, Moscow,Russia
[2]Sodankylä Geophysical Observatory, Sodankylä, Finland

**Correspondence:** N. Yagova(nyagova@ifz.ru)

**Abstract.** Using data of the ionosonde in Sodankylä, (SOD, 67°N, 27°E, Finland), parameters of variations of foF2 critical frequency in the Pc5/Pi3 ($1 - 5$ mHz) frequency range are studied. For that, a technique of automatic detection of critical frequency from an ionogram is developed. The variations of foF2 are compared with the Pc5/Pi3 geomagnetic pulsations on the ground and in the magnetosphere. The variations of foF2 are in the majority of cases decoupled from the Pc5/Pi3 on the ground. Meanwhile, the analysis of geomagnetic and foF2 variations at SOD show intervals with noticeable coherence for both horizontal components. These coherent pulsations are predominantly registered in the afternoon MLT sector. Statistically, their spectral content, polarization and spatial distribution differs from those of background variations. Coherent pulsation tend to occur under moderate geomagnetic and auroral activity, SW speed, and dynamic pressure fluctuations. The fraction of coherent geomagnetic and foF2 pulsations is higher for the geomagnetic pulsations registered in the magnetosphere, than on the ground.

10 *Copyright statement.* TEXT

## 1 Introduction

Modulation of ionospheric parameters by Pc5 pulsations was reported by Pilipenko et al. (2014a, b) Majority of publications report on the radar observation (Mager et al., 2015; James et al., 2016), i.e. the observations variations of electron concentration at certain altitude in the ionosphere. Observations of pulsations in the total electron content (TEC) are rather rare
15 (Pilipenko et al., 2014a, b; Watson et al., 2015). Watson et al. (2015) reported on TEC variations measured by GPS at Pc5-6 frequencies. The large-amplitude TEC variations were associated with mainly compressional mode of MHD wave in the magnetosphere. The pulsations were also seen in magnetic field on the ground with two spectral peaks at about 0.9 mHz and 3.3 mHz. The event was observed in the afternoon MLT sector after a steep increase of SW dynamic pressure up to almost 20 nPa.

An intriguing effect of double Pc5 frequency in fluctuations of ionospheric parameters was shown by Kozyreva et al. (2019).
20 However, the effect of ionosphere heating by an intense MHD wave, found by Pilipenko et al. (2014b) at the recovery phase of the magnetic storm is possible only for extremely high Pc5 amplitudes. On the contrary, a role of MHD waves with moderate amplitudes in variations of foF2 critical frequency has not been studied in details.





Our study is aimed on variations of foF2 critical frequency at Pc5/Pi3 frequencies and geomagnetic pulsations in the same frequency range on the ground and in the magnetosphere, both for individual events and statistically.

## 2   Data of observations and their processing

### 2.1   Data

The Sodankylä Geophysical Observatory (SOD) ionosonde is located at geographic coordinates 67.3 N, 26.7 E. It makes an ionogram once a minute. A detailed description of observations can be found in (Kozlovsky et al., 2013). SOD magnetometer is a part of IMAGE magnetometer network Taskanen (2009), and three components of the geomagnetic field are available with 10 s sampling rate. For the analysis of Pc5 spatial distribution, we also use the data of another IMAGE station MAS. Station information is summarized in Table 1. For the analysis of geomagnetic pulsations in the magnetosphere, we use data of the flux-gate magnetometers at THEMIS satellites (Auster et al., 2008). For individual events, pressure pulsations measured at THEMIS have been used, as well. The data of PLASMA experiment at THEMIS can be found in (McFadden et al., 2008a, b).

### 2.2   foF2 automatic detection from ionograms

Although visual detection of a critical frequency from an ionogram with clearly expressed layers is not difficult, an automatic detection method is necessary for an analysis of high frequency variations of critical frequencies. The difficulties of this procedure are caused by different intensity of the reflected signal, different contrast between reflection maximum and the background, occurrence of sporadic layers, man-made interference, etc. This makes routine techniques, based on the estimates at a given frequency-altitude windows non-stable even in cases, when visual detection is possible.

Below, we present a method based on the approximation of the reflection boundary in a wide range of altitudes to suppress the influence of local gaps and peaks in reflection.

The high frequency reflection boundary in such a presentation is characterized by almost linear growth of frequency at low altitudes, then growth becomes slower, and finally it saturates at critical frequency. We approximate this dependence by a Lorentsian type function

$$f(h) = f_1 + \Delta f \frac{k(h - h_1)^\alpha}{k(h - h_1)^\alpha + 1} \tag{1}$$

Starting altitude is taken $h_1 = 235$ km Coefficients $f_1$, $\Delta f = f_2 - f_1$, $k$, and $\alpha$ are found as a result of fitting procedure, described below. The boundary is determined as a line where the following two conditions are fulfilled:

- Signal power $P$ at the boundary should be high

- Amplitude ratio $R$ of the signal power at the boundary line to the power above it should also be high

As four fitting factors are used, a 9-point iteration procedure is organized and a parameter $P_t$, determined from $P_t^2 = cP^2 + (1-c)R^2$ is maximized over the 4-D cross $P_t(x_0, x_0 - \Delta x_i, x_0 + \Delta x_i)$, where $x$ is a point in the space of parameters,





and $i$ is a parameter number. The initial approximation is taken from the database created manually for several typical types of $f(h)$ dependence. foF2 is then determined as a value of (1) in the altitude region with weak dependence $f(h)$. The other requirement is a continuity of time dependence $f(t)$. The threshold value for the time derivative of foF2 is estimated from

the variance of an interval of length $t_1$. For $t > t_1$, the set of parameters calculated at the previous step is taken as the initial approximation. If an iteration procedure gives a value of foF2 with the difference from the previous values exceeding the threshold value, the other initial approximation from the database is taken, and the procedure is repeated. If all the initial approximations give a value, outstanding far from the previous ones, this point is excluded, and the iteration procedure is started from the next time instant. The results for selected days are given in the (APPENDIX). Examples of approximation

curves are given in Figure 1 for 3 ionograms, registered on day 2014 297. The continuity condition allows to suppress the influence of additional reflection maxima and bifurcations. The list of the days and time intervals selected with the automatic detection procedure and tested visually for each tenth point, ASCII files and pictures of foF2 time variations for the selected intervals are available in supplementary files. The selected intervals form the database for the analysis.

  An example of diurnal variations of critical frequency obtained with the technique described is given in Figure 2 for the

DAY 2015 003. Note, that the ionograms are rotated by $90°$ in respect to usual $f - H$ presentation.

### 2.3 Pre-processing, statistical and spectral analysis

For the ground pulsations, two horizontal components are analyzed, while for the pulsations in the magnetosphere we use three components in the local magnetic field-aligned coordinate system. The $B_\parallel$ component is directed along the main magnetic field $\mathbf{B}$, $B_\rho$ is the component, transversal to $\mathbf{B}$ and lying in the plane including $\mathbf{B}$ and the Earth center, and the $B_\varphi$ component is

normal to both $B_\parallel$ and $B_\rho$, and its direction is selected to make $B_\parallel$, $B_\rho$, and $B_\varphi$ right-side triple. We use the notation $b$ for the pulsation magnetic field to discriminate it from the main magnetic field $B$ and we denote pulsations of foF2 and $P$ as $\Delta$foF2 and $\Delta P$, respectively.

  Statistical analysis includes distribution over MLT, parameters of pulsations and the space weather during the interval analyzed and preceding it. We studied Dst and AE geomagnetic indexes, vertical component of the interplanetary magnetic field

IMF $B_Z$, solar wind velocity $V$ and dynamic pressure $P$ and the maximal amplitude $P$ fluctuations.

  The method, described in the previous section, allows to get foF2 with the time resolution, enough for spectral estimates and comparison with the geomagnetic pulsations based on cross-spectral analysis. For that, power spectral density (PSD) is estimated with a Blackman-Tukey method (Kay, 1988) in a sliding 64 points window with 10 min shift between adjacent intervals. Cross-spectra are calculated for foF2 variations, on one hand, components of the geomagnetic field pulsations, on the

other hand. For the intervals with high spectral coherence $\gamma^2$, phase difference $\Delta\varphi$ is estimated.





## 3 Results

### 3.1 foF2 variations and geomagnetic pulsations at SOD

#### 3.1.1 Examples

We present two examples of foF2 and geomagnetic variations simultaneously recorded at SOD. Variations of geomagnetic
field components and foF2 at SOD for the event 1 (day 2015 070) are presented in Figure 3. Peak-to-peak amplitudes of
geomagnetic field and foF2 are about $10$ nT and $0.08$ MHz, respectively. PSD for both geomagnetic and foF2 variations,
spectral coherence and phase difference are presented in Figure 4. The PSD spectrum of geomagnetic pulsations has two broad
maxima at $f_1 = 2.3$ and $f_2 = 3.2$ mHz. The spectrum of foF2 variations has a maximum at a frequency $f = 3.2$ mHz, i.e. at a
$f_2$ frequency. Meanwhile, spectral coherence is high ($\gamma^2 > 0.5$) at $f < 2$ mHz and near the $f_2$ frequency.

Figure 5 illustrates the space weather conditions. The start point of the interval is taken as zero of time axis $\tau$ at panels (a-e)
of Figure 5. It is seen from the Figure, that geomagnetic conditions were quiet and no magnetic storms occurred during at
least four days before the event, Dst>-20 nT (Figure 5a). However, the auroral activity was essential and maximal AE reached
$500$ nT (Figure 5b). This activation followed the interval of negative $B_Z$ with variations of almost $20$ nT amplitude (Figure
5d). For this event, SW speed $V$ was about $400$ km/s (Figure 5c), SW dynamic pressure $P \approx 4$ nPa (Figure 5e). $P$ fluctuations
are shown in more details in Figure 5f. Their amplitude was about $0.7$ nPa and their apparent period was about $5$ minutes. This
corresponds to frequency $f = 3.3$ mHz, i.e. it approximately agrees with the $f_2$ frequency of pulsations at SOD.

The results for event 2 (Day 2015 192) is presented in Figures 6 and 7, which have the same format, as Figures 3 and 4. Peak-
to-peak amplitudes of the geomagnetic and foF2 pulsations are about $80$ nT and $0.08$ MHZ, respectively. A clear maximum
at $f_1 \approx 2.5$ mHz is seen in both geomagnetic and foF2 PSD spectra (Figure 7a). At the second frequency $f_2 \approx 3.5$ mHz a
maximum is seen only in foF2 variations, while in the geomagnetic pulsations this frequency is marked only as a plateau in the
PSD spectrum. However, both spectral maxima are seen clearly in the coherence spectrum (Figure 7b), and the phase difference
is different for these two frequencies (Figure 7c).

Space weather conditions for this event are summarized in Figure 8, which has the same format as Figure 5. No geomagnetic
storms were registered during last 4 days before this event, as Dst exceeds $= -30$ nT throughout the interval (Figure 8a).
Meanwhile auroral activity is high: two auroral activations are seen at $\tau = -8$ and $-4$ hours with maximal AE= $1300$ nT and
($700$ nT), respectively (Figure 8b). The first activation developed after 2 hour interval of negative $B_z$, while the second one
corresponds to $B_z$ turn from $-10$ to almost $+15$ nT (Figure 8d). For this event, $V$ is about $600$ km/s (Figure 8c), maximal
$P$ was $\approx 9$ nPa and then dropped to $5$ nPa and slowly decrease to about $3$ nPa (Figure 8e). Amplitude of $P$ fluctuations
exceeded $1$ nPa and their apparent period was about $7 - 8$ minutes (Figure 8f). The frequency of $P$ pulsations is $3.3$ mHz, i.e.
it approximately corresponds to $f_2$ frequency in foF2 variations, registered at SOD.





### 3.1.2 Statistics

A MLT distribution of occurrence of the foF2 variations is shown in Figure 9. One can see from the Figure, that Pc5/Pi3 variations of foF2 are predominantly registered in the post-noon MLT sector with maximal probability at MLT 12-15.

Figure 10a shows frequency distributions of geomagnetic and foF2 pulsations at SOD. $f_1$ is a frequency of the first spectral
maximum in the range from 1.5 to 5.5 mHz. The frequency distribution of foF2 fluctuations is enriched with frequencies ($f_1 > 3.7$ mHz) in comparison with the distribution of the geomagnetic pulsations. The distribution of Pc5/Pi3 intervals over $foF2 - b$ spectral coherence at SOD are shown in Figure 10b for two horizontal components. For both components, spectral coherence $\gamma^2 < 0.375$ dominates. The fraction of $\gamma^2 \geq 0.375$ intervals is $1/6$ and $1/8$ for $b_X$ and $b_Y$ components, respectively, the fraction of $\gamma^2 \geq 0.5$ is less than 3% for both components.

This means, that in majority cases, Pc5/Pi3 geomagnetic pulsations and variations of foF2 critical frequency at the same point are decoupled. This effect can be seen from both different spectral content and low spectral coherence of magnetic and foF2 fluctuations.

However, the coherent foF2 and geomagnetic pulsations do exist, and a question arises about the pulsation properties and external parameters, favorable for their occurrence. To answer this question, the geomagnetic pulsations at SOD for which
$b_X - foF2$ coherence is high ($\gamma^2 > 0.5$) are compared with all the Pc5s registered at SOD during 21 months from April of 2014 till the end of 2015. To avoid the influence of different seasonal and diurnal variations of the selected and other pulsations (referred to as "background"), the statistics of background pulsations is calculated with the weight functions calculated from the seasonal and diurnal variations of coherent pulsations. Figure 11 illustrates the difference between coherent and background pulsations for three parameters: $P_f^{bx}$ (Figure 11a), PSD ratio $R_{XY} = P_f^{bx}/P_f^{by}$ (Figure 11b), and the $b_X$ PSD ratio along a
magnetic meridian $R_\Phi = P_f^{bx}(\Phi)/P_f^{bx}(\Phi + \Delta\Phi)$ (Figure 11c). The latter is calculated for SOD-MAS station pair (MAS station is located nearly at the same magnetic meridian, but it is shifted in $2°$ northward). $P_f^{bx}$ for coherent pulsations is enriched with frequencies $f > 2$ mHz in comparison with the background pulsations. In this frequency band, $R_{XY}$ also increases and $R_\Phi$ demonstrates a non-monotonous dependence on frequency with minimum at $f = 2.7$ mHz and growth at $f \geq 3$ mHz. These features are only weakly seen in $R_\Phi(f)$ dependence for the background pulsations.

To understand, what space weather conditions are favorable for generation of coherent $b_X - foF2$ pulsations, we compare the geomagnetic indexes and SW/IMF conditions for intervals when coherent $b_X - foF2$ and background pulsations were registered. The influence of seasonal and diurnal variation was eliminated in the same manner, as for pulsation parameters. We use for the analysis the 4-day minimum Dst and 6-hour maximal AE, as Pc5 amplitudes are maximal at recovery phase of geomagnetic storms (Posch et al., 2003), and auroral substorms are followed by Pi3 pulsations (Kleimenova et al., 2002)
and Pc5 waves with high azimuthal and intermediate wavenumbers (Zolotukhina et al., 2008; Mager et al., 2019). The results for Dst and AE indexes are summarized in Figure 12. Coherent pulsations tend to occur under moderate geomagnetic and auroral activity. The most favorable Dst interval is from $-100$ to $-50$ nT (Figure 12a), and for AE index it is from 250 to 500 nT (Figure 12b). Under highly disturbed conditions, probability to register coherent foF2-$b_X$ pulsations vanishes. This result naturally follows from the condition of existence of clear layer structure, necessary for the pulsation detection procedure.





During geomagnetic storms detection of the foF2 variations is often impossible because of enhanced ionization in the lower ionospheric layers (E and/or D).

Occurrence and parameters of high-latitude Pc5s are controlled by interplanetary parameters, especially IMF $B_Z$ component, variations of solar wind dynamic pressure $P$, and solar wind velocity $V$ (Baker et al., 2003). Distributions of these three parameters for coherent and background events are presented in Figure 13. We have taken 3-hour mean values of $B_Z$ and $V$ and 3-hour maximal value of $\Delta P$. Figure 13a shows that coherent events tend to occur during positive $B_Z$ intervals. This result agrees with moderate geomagnetic and auroral activity, favorable for coherent foF2-$b$ pulsations. Figures 13 (b) and (c) show that, comparing to background fluctuations, the selected coherent events tended to occur during somewhat higher SW speed and higher amplitudes of SW pressure fluctuations.

### 3.2 foF2 variations at SOD and pulsations in the magnetosphere

Although variations of foF2 are in the majority cases decoupled from ground Pc5/Pi3 at the same site, magnetospheric properties of the coherent waves, which contribute to high coherence tails in the distribution shown in Fig. 10b, are important to understand their origin. During the years 2014-2015, THEMIS D and E satellites' orbits crossed the magnetosphere, and for several events, when fluctuations of foF2 in the Pc5/Pi3 frequency range were registered, the data of the magnetic field and plasma parameters at one or several THEMIS satellites were also available. We have made spectral estimates for variations of these parameters at THEMIS with the same technique, as for ground variations.

#### 3.2.1 Examples

Pulsations in Pc5/Pi3 frequency range were registered simultaneously in foF2 at SOD and in the magnetic field at THEMIS on day 2014 344 (Event 3). At 11 UT, THEMIS-D was in the plasma sheet at the radial distance about $12\ R_E$, and the CGM coordinates of its Northern footprint were $\Phi = 65°, \Lambda = 212°$, i.e. it was at $L \approx 7$, against $L \approx 5$ at SOD, it was shifted at almost 7 hours in MLT. Coordinates of THEMIS satellites and their footprints for the events 3 and 4 are summarized in Table 2.

The time variations foF2 at SOD and 3 components of the magnetic field at THEMIS-D are shown in Figure 14 for the 1-hour interval, starting at 10:50 UT. The apparent period of variations is about 6 minutes, and peak-to-peak amplitudes of geomagnetic pulsations are about $15$ and $3$ nT at SOD and THEMIS-D, respectively. Peak-to-peak amplitude of foF2 fluctuations is about $0.1$ MHz.

PSD spectra, spectral coherence and phase difference for this event are shown in Figure 15. The main spectral maximum is found at $f_C = 2.7$ mHz in spectra of foF2, meridional component at SOD and field-aligned ($b_\parallel$) component at THEMIS-D. The nearest maxima in spectra of transversal components at THEMIS-D are shifted to lower frequency ($f_A = 2.4$ mHz), and minor maxima are seen in foF2 and magnetic field fluctuations at frequencies below 2 and higher than 3 mHz. Two main maxima in spectral coherence are found at $f_{\gamma\parallel} = 3$ mHz for $B_x$ at SOD and $B_\parallel$ at THEMIS-D, and $f_{\gamma\perp} = 2.2$ mHz for transversal components at THEMIS-D. The phase difference at the frequency of the main coherence maximum $\Delta\varphi(f_{\gamma\parallel}) = 30, 45°$ for





THEMIS-D $b_{\|}$ and SOD $b_X$, respectively. For the second coherence maximum $\Delta\varphi(f_{\gamma\perp}) = 45, -135°$ for THEMIS-D $b_\phi$ and $b_\rho$, respectively.

Space weather conditions for this event are summarized in Figure 16. Dst exceeded $-30$ nT (Figure 16a). Two auroral activations started at $\tau = -12$ hours with maximal AE= 350 nT and at $\tau = -2$ hours with maximal AE= 500 nT (Figure 16b). The first activation occurred during predominantly negative and highly variable $B_Z$, while the second one developed after about two hours of weakly negative $B_Z > -2.5$ nT (Figure 16d). SW speed varied from about 500 km/s to 450 km/s during last hours before the interval of analysis (Figure 16c) and SW dynamic pressure was fluctuating around $P \approx 2$ nPa (Figure 16e). During the interval of analysis, SW dynamic pressure suffered fluctuations with peak-to-peak amplitude about 1 nPa and main period about 3 minutes (Figure 16f). This period corresponds to $f \approx 5.5$ mHz, i.e. it is about double frequency of the geomagnetic pulsations, registered during the same interval in the magnetosphere and on the ground.

On Day 2015 265 (event 4), Pc5/Pi3 fluctuations were also registered simultaneously in magnetic field at THEMIS-D and in foF2 at SOD. At 14 UT, THEMIS-D was in the night magnetosphere at the radial distance about 12 $R_E$, and the CGM coordinates of its Northern footprint was at $\Phi = 69, \Lambda = 294$, i.e. at $L \approx 8$ in the MLT sector, opposite to SOD.

Time series of magnetic field components and foF2 for the interval starting at 13:50 UT are shown in Figure 17 in the same format, as for the previous event. Peak-to-peak amplitude of $b_\varphi$ variations at THEMIS-D is about 5 nT, and for radial and field aligned components it is $1.5 - 2$ times less. On the ground at SOD, maximal peak-to-peak amplitude reached 10 nT and 0.05 MHz in $b_X$ and foF2 variations, respectively. PSD spectra, spectral coherence and phase difference for this event are shown in Figure 18. Both PSD spectra (Figure 18a) and spectral coherence (Figure 18b) demonstrate a closer association between foF2 variations and the magnetic field pulsations at THEMIS-D, than at SOD. Indeed, the main PSD and coherence maxima are found at the same frequency $f_C = 3$ mHz and it is seen in all three THEMIS-D components, but the coherence for transversal components is higher. Variations of $b_\phi$ component and foF2 are almost in-phase at this frequency, and the phase difference for two other components $\Delta\varphi(f_C) = -110, 20°$ for THEMIS-D $b_{\|}$ and $b_\rho$, respectively.

Space weather conditions for this event are illustrated in Figure 19. Pulsation developed on the recovery phase of a moderate magnetic storm with minimal Dst= $-76$ nT at $\tau = -52$ (Day 263, Figure 19a). Auroral activity also was high with maximal AE$\approx$ 1000 nT at $\tau = -6$ hours (Figure 19b). This activation followed $B_Z$ jump from $-3.5$ to 5.5 nT at $\tau = -8$hours (Figure 19d). The SW speed reached 600 km/s Figure 19c), while $P$ was about 2.5 nPa (Figure 19e). The amplitude of $P$ fluctuations during the last hour before the interval of analysis was about 0.7 nPa (Figure 19f). This event occurred at the most disturbed background among all the cases analyzed.

### 3.2.2 Statistics

The distribution of coherence of foF2 variations at SOD and two components of magnetic field registered by the THEMIS-D satellite in the magnetosphere are shown in Figure 20. The Figure shows, that the distribution of $\gamma^2$ for foF2 - $b_{\|}$ is enriched with $\gamma^2 > 0.375$ values in comparison with the similar distribution at SOD (Fig 10b). The fraction of $\gamma^2 > 0.375$ intervals for the transversal components at THEMIS-D is nearly the same as for X-component on the ground.





## 4 Discussion

We have found that in majority cases, variations of foF2 in the Pc5/Pi3 frequency range are decoupled from ground Pc5 at the same site. However, coherent variations occur sometime, preferably in the afternoon MLT sector. The fraction of coherent pulsations is higher for the variations of foF2 at SOD and the field aligned component of the magnetic field in the magnetosphere (recorded by the THEMIS-D satellite).

Coherent foF2-b pulsations develop mostly under moderately disturbed geomagnetic conditions. A better correspondence is found between $foF2$ variations and pulsations of filed-aligned component in the magnetosphere, than that for transversal components in the magnetosphere and both horizontal components on the ground. This effect can result from the effective screening of waves with a small transversal scale (Kokubun et al., 1989). The direct measurements of ULF wavenumbers in the ionosphere carried out by Baddeley et al. (2005) really showed high $m$ values, different for two classes of pulsations.

Below we present a more detailed analysis of wave properties for the event 4. Figure 21 presents magnetograms for field-aligned component at THEMIS-D and E satellites. Although the distance between the two satellites is only $0.5\ R_e$, and the distance between their footprints does not exceed $0.5°$ both in latitude and in longitude, the phase difference is about $\pi/4$.

For this event, plasma pressure data are also available. The pulsation of field-aligned magnetic field and ion pressure at THEMIS-D together with foF2 variations at SOD are shown in Figure 22. Time series for two pairs of variables $b_{\parallel} - \Delta P$ and $foF2^2 - \Delta P$ are shown in Figure 22(a-b). We use the square of foF2, in this Figure, as it is proportional to the electron concentration, and the correspondence with variations of ion pressure is seen more explicitly in this parameter. Variations of magnetic field and pressure are almost counter-phase (Figure 22a), while variations of pressure and $foF2^2$ are almost in phase (Figure 22b). Normalized PSD spectra for all three parameters, spectral coherence and phase difference are presented in Figure22(c-e). We see two spectral maxima in $\Delta P$ PSD spectrum at $f_1 \approx 1.5$ mHz and $f_2 \approx 3$ mHz (Figure 22c). Both maxima are seen in $foF2^2 - \Delta P$ coherence spectra, as well (Figure 22d) . The phase difference at the lower frequency is almost zero and it is about $-60°$ at the higher frequency (Figure 22e) . The maximum near the $f_2$ frequency is also seen in $b_{\parallel}$ PSD and $b_{\parallel} - \Delta P$ coherence spectra. The phase difference at this frequency for $b_{\parallel} - \Delta P\ \Delta\varphi \approx 180°$ (Figure 22e) . The 1-hour mean of the magnetic absolute value at THEMIS-D is $|B| \approx 30$ nT, and the amplitude of $b_{\parallel}$ pulsation is about $1.5\ nT$, i.e. 5% of the undisturbed value. The pressure mean is $|P| \approx 770$ eV/$cm^3$, and the amplitude of $\Delta P$ pulsations is about 70 eV/$cm^3$, i.e. it is about 10% of the undisturbed value and the balance of plasma and magnetic pressure is fulfilled in this pulsation in the magnetosphere. $\Delta N_e/N_e$ value in the ionosphere, as estimated from foF2 is about 1%, and at SOD, $\Delta B/B$ on the ground is an order of magnitude lower. Thus for the events studied, $\Delta B/B$ amplitude ratio on the ground ratio is small in comparison with $\Delta N/N \approx 2\Delta F/F$ in the ionosphere, which, in its turn is less than $\Delta B/B$ in the magnetosphere. The wave in the magnetosphere is characterized by 10% modulation of both magnetic and plasma pressure.

Amplitude of SW dynamic pressure fluctuations show an association with occurrence of coherent $foF2 - B$ pulsations. Some of individual cases (Figures 5, 8) show also close periods of $P$ and $foF2 - B$ pulsations. However, the influence of $P$ transient and quasi-periodical variations on $foF2$ variations and and its possible relation to high latitude Pc5/Pi3s (Kepko et al., 2002; Kim et al., 2002; Yagova et al., 2007) are issues for a special study.





For events with low amplitudes of geomagnetic pulsations on the ground, the particle flux and/or pressure modulation by
a compressional wave seems the most probable source of foF2 variations. Meanwhile, in the events when ground Pc5s are
coherent with foF2 variations, the magnetic pulsations show typical features of the shear Alfven resonance in spectral content,
polarization, and amplitude distribution along the meridian. A similar result follows from the comprehensive statistical analysis
of correspondence between geomagnetic and Cosmic Noise Adsorbtion (CNA) pulsations (Spanswick et al., 2005), who found
that geomagnetic pulsations with FLR features demonstrate a better correspondence with CNA pulsations than non-FLR Pc5s.
However, physical reasons for our and (Spanswick et al., 2005) results may be different, because of different particle en-
ergies and Pc5 types. A detailed case study of magnetic field and electron flux pulsations at four Cluster satellites located at
different L-shells in the magnetosphere and geomagnetic and CNA pulsations on the ground (Motoba et al., 2013) showed
rather complicated space distribution and time variation of geomagnetic and electron pulsations and their inter-relation. The
picture changed dramatically within 30-40 minutes, and the pulsation in space was, probably, a mix of compressional and
Alfven modes. The authors found that the amplitude of compressional mode was critical for effective modulation of electron
flux, but the contribution of shear Alfven resonance was also non-negligible. Our analysis has also shown typical features
of Alfven resonance (Baransky et al., 1995) in coherent $foF2 - B_X$ pulsations in comparison with typical afternoon Pc5s at
SOD.

## 5  Conclusions

For the first time, a statistical study of foF2 variations in Pc5/Pi3 range and their relation to geomagnetic pulsation on the
ground and in the magnetosphere is carried out. It is shown that not only storm-time Pc5s can modulate the ionosphere foF2,
but but also non-storm pulsations with moderate amplitudes can modulate foF2. It is important, because in such conditions, F2
layer is not blanketed by lower layers. Case studies show that $\Delta B/B$ to $\Delta N/N$ ratio varies in wide range. Statistical analysis
shows that coherent $foF2 - b_x$ pulsations at SOD demonstrate clear Alfven resonance features. On the other hand, in the
magnetosphere, a higher coherence between foF2 and magnetic pulsations is found in field aligned component, while spectral
coherence for the transversal components in the magnetosphere is close to that on the ground.

The above analysis found a dependence of the occurrence of coherent foF2-$b$ pulsations on the level od SW pressure fluc-
tuations. This effect can be related to global compressional mode, generated by SW pressure fluctuations. In contrast to them,
Alfven waves can modulate upper ionosphere under very high amplitudes only (Pilipenko et al., 2014b).

*Sample availability.* All the intervals used in the analysis are visually checked (for each 10-th point) and presented in the table file. The foF2
values, obtained with Eq.(1) for all the intervals analyzed, are available both as jpeg figures and ASCII files. A file name has a structure
SOD-YYYY-DDD-foF2, where YYYY is a year and DDD is a day number. Each ASCII file contains two columns:

   1. time (seconds) from 00:00 UT

   2. foF2 (MHz).





*Author contributions.* N. Yagova: the approximation algorithm, cross-spectral and statistical analysis; A. Kozlovsky: pre-processing and visual check of ionosonde data; E. Fedorov: interpretation of results, analysis of wave parameters; O. Kozyreva: selection of events, algorithms and codes for visualization; all the authors: MS preparation

*Competing interests.* The authors have no competing interests

*Acknowledgements.* We thank SGO (http://www.sgo.fi/) for SOD magnetometer and ionosonde data, Finnish Meteorological Institute for
MAS magnetometer data, CDAWEB (https://cdaweb.gsfc.nasa.gov) for THEMIS and OMNI data, and Wolld data center Kyoro (http://wdc.kugi.kyoto-u.ac.jp/index.html) for AE and Dst indexes. The study was supported by the Academy of Finland grants 298578 and 310348 (OK) and state contract with IPE (NY, EF). Useful discussions with V.A. Pilipenko are appreciated.



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



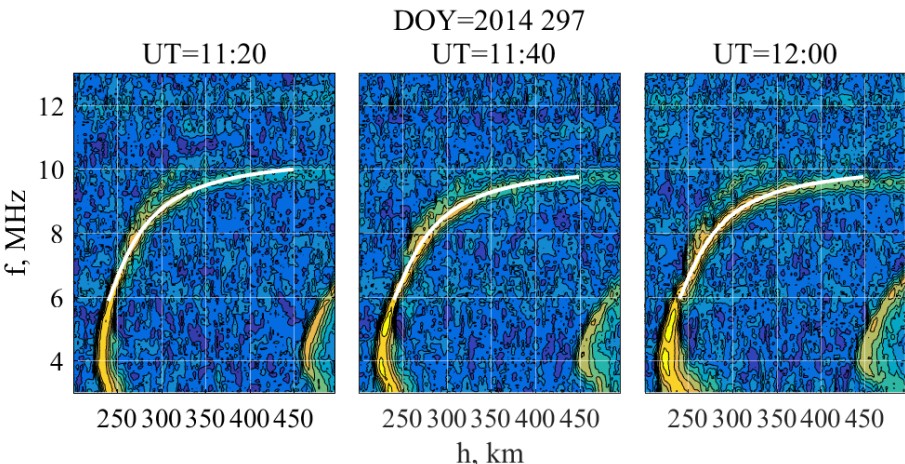

**Figure 1.** Examples of approximations of f(h) dependence with eq. (1)





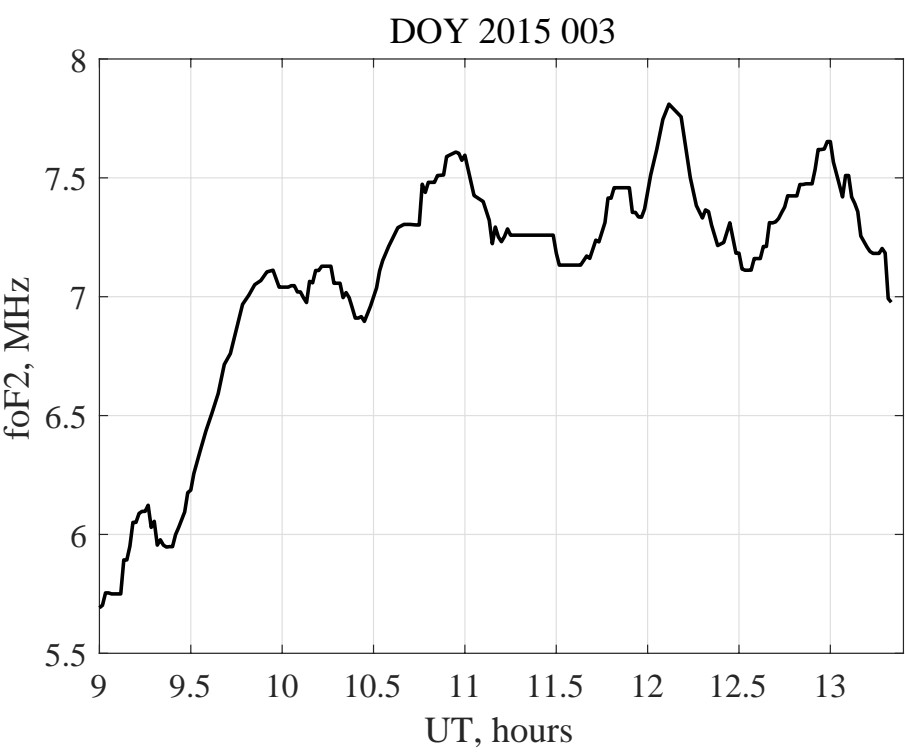

**Figure 2.** Variation of foF2 frequency during 4.5 hours on day 2015 003 , obtained with eq. (1)

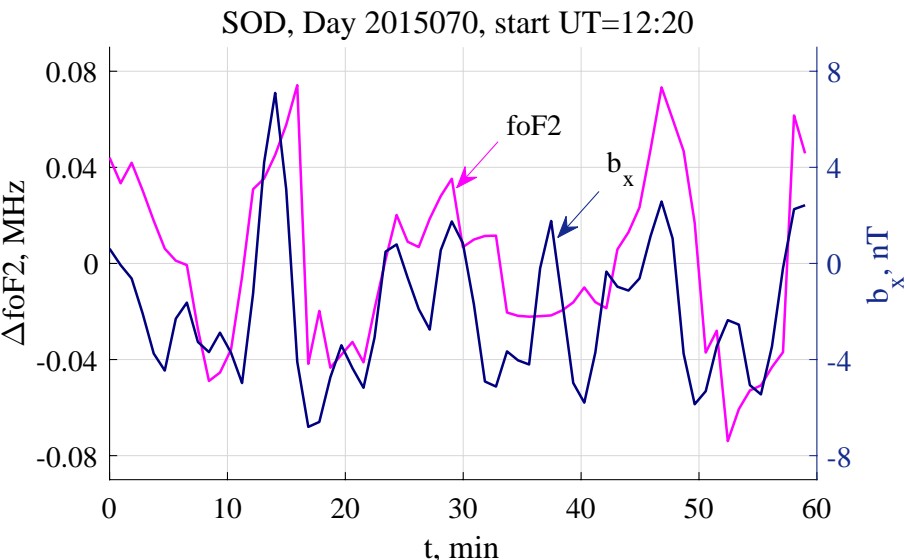

**Figure 3.** Variation of foF2 and geomagnetic pulsations at SOD during event 1

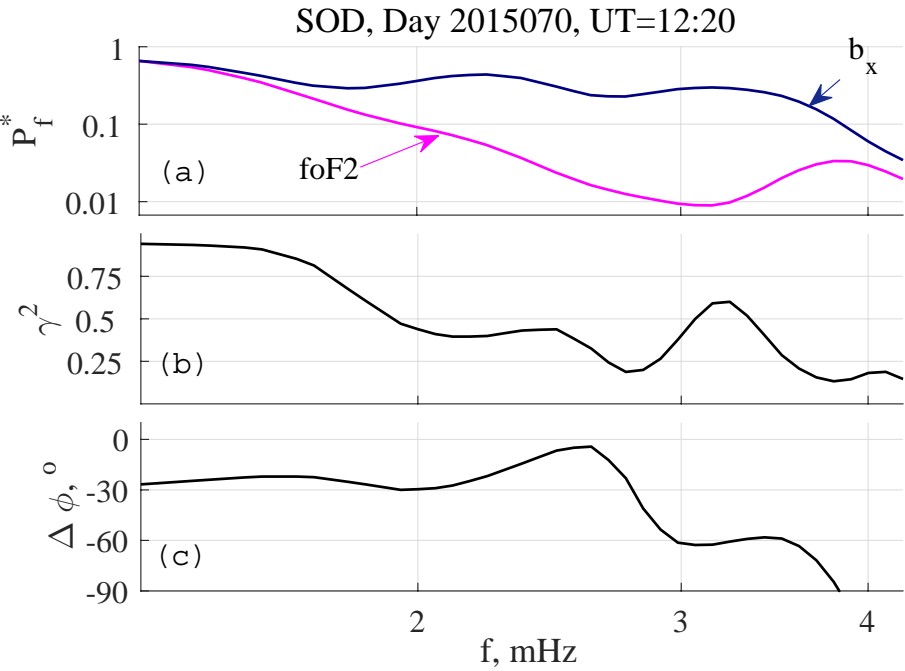

**Figure 4.** Spectral parameters for the event 1: (a) normalized PSD spectra of foF2 and $b_X$ pulsations, (b) spectral coherence between foF2 and $b_X$; (c) phase difference

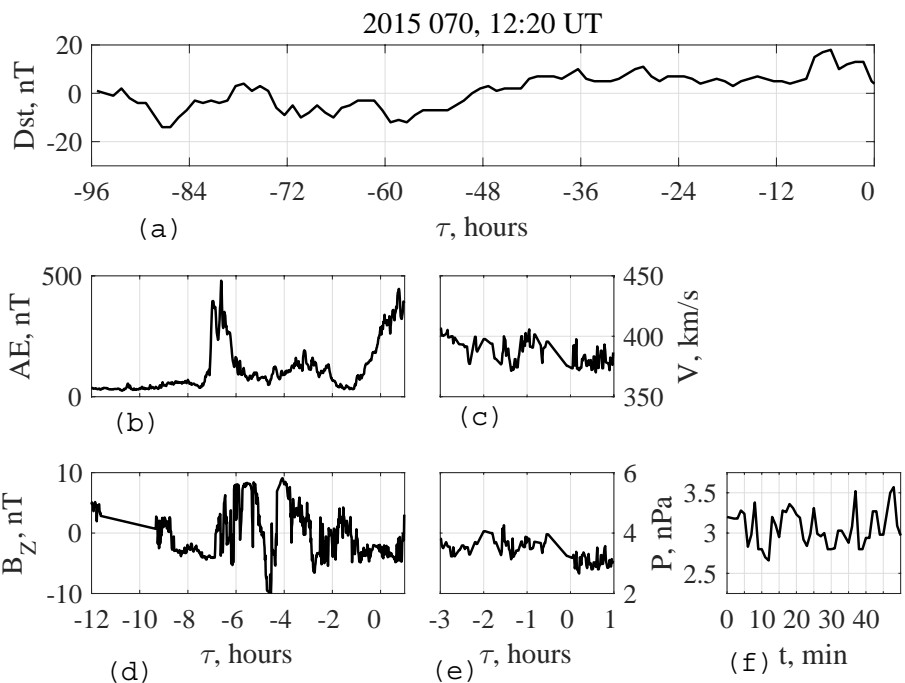

**Figure 5.** Space weather conditions for the event 1. (a) Dst index during last four days; (b) AE index during the interval and 12 hours before; (c) SW speed during the interval and 3 hours before; (d) IMF $B_Z$ during the interval and 12 hours before; (e) SW dynamic pressure during the interval and 3 hours before; (f) details of SW dynamic pressure fluctuations during the interval.

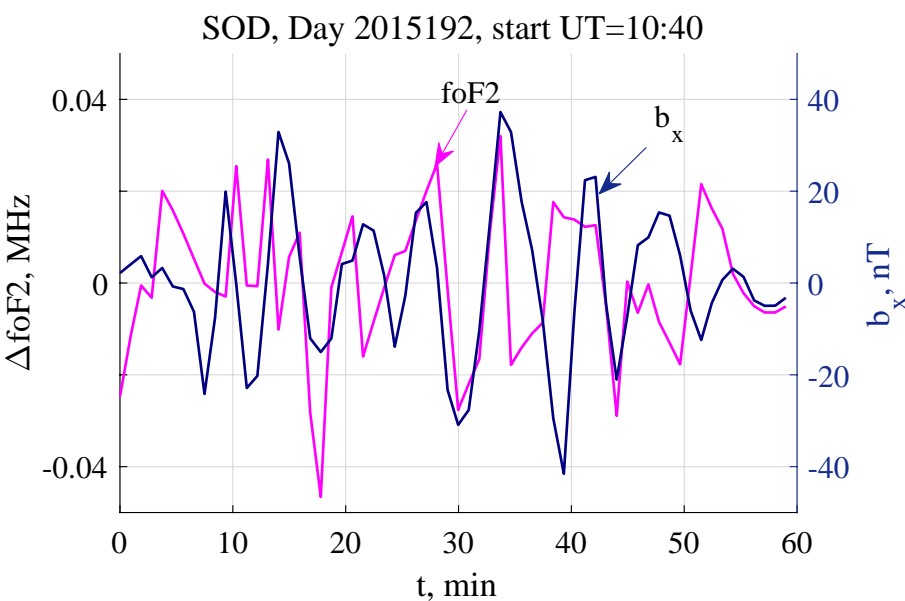

**Figure 6.** Variation of foF2 and geomagnetic pulsations at SOD during event 2

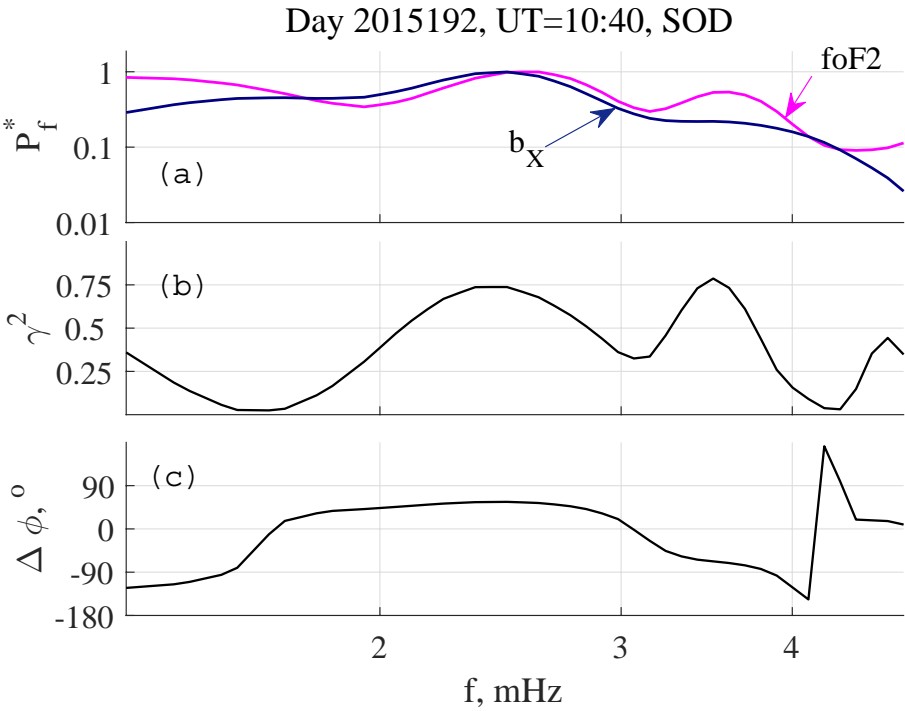

**Figure 7.** Spectral parameters for the event 2: (a) normalized PSD spectra of foF2 and $b_X$ pulsations, (b) spectral coherence between foF2 and $b_X$; (c) phase difference.

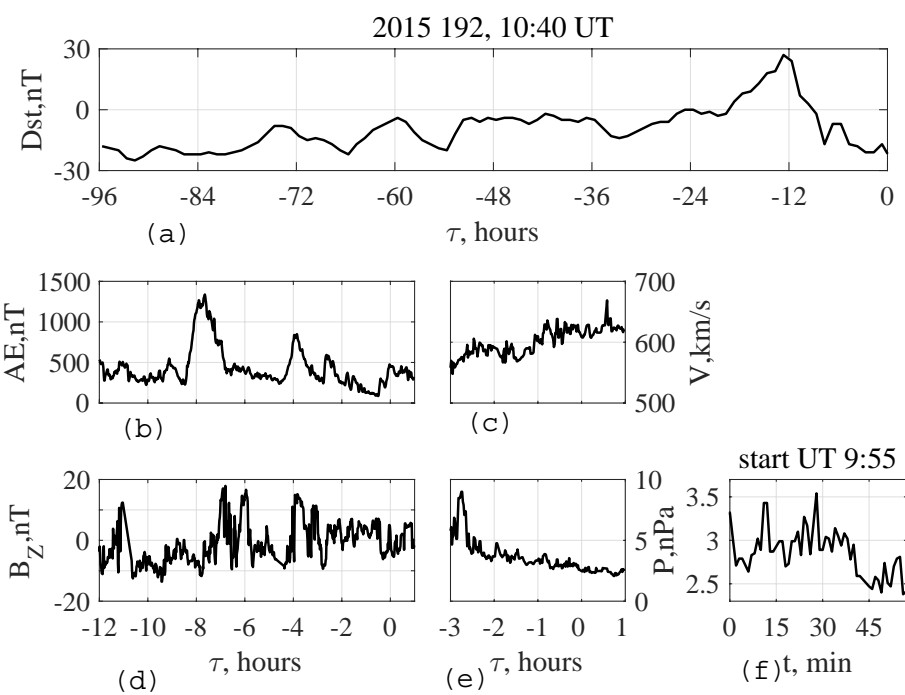

**Figure 8.** Space weather conditions for the event 2. (a) Dst index during last four days; (b) AE index during the interval and 12 hours before; (c) SW speed during the interval and 3 hours before; (d) IMF $B_Z$ during the interval and 12 hours before; (e) SW dynamic pressure during the interval and 3 hours before; (f) details of SW dynamic pressure fluctuations during the last hour before the interval.

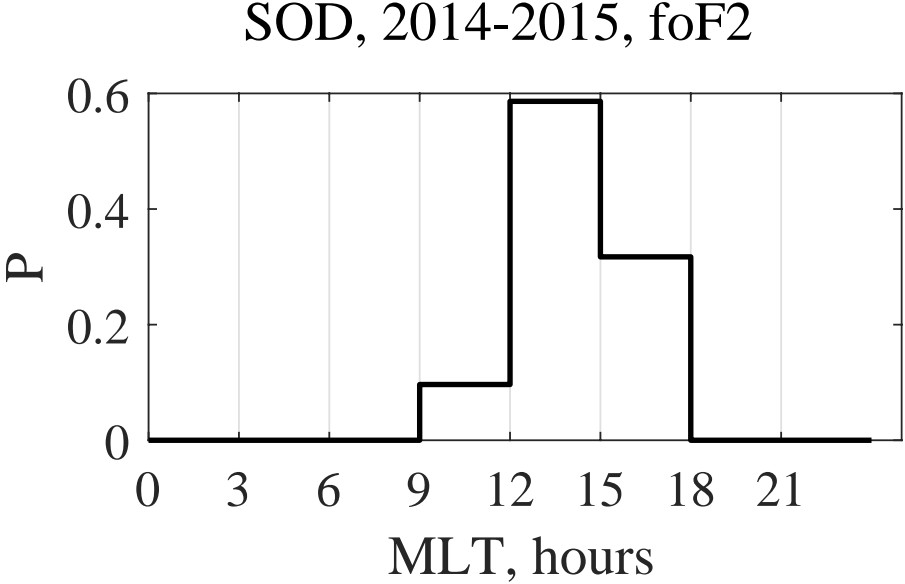

**Figure 9.** Averaged over 2 years MLT distribution of occurrence of foF2 fluctuations in $1 - 5$ mHz frequency band.



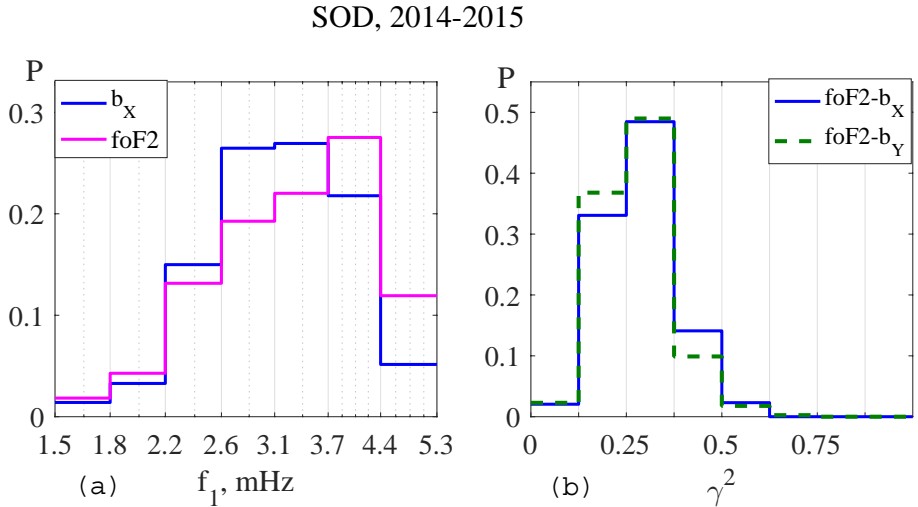

**Figure 10.** Averaged over 2 years distributions parameters of foF2 fluctuations and Pc5/Pi3 pulsations at SOD: (a) frequency of the first spectral maximum of $b_X$ (blue) and foF2 (magenta); (b) spectral coherence of foF2-$b_X$ (blue) and foF2-$b_Y$ (green).



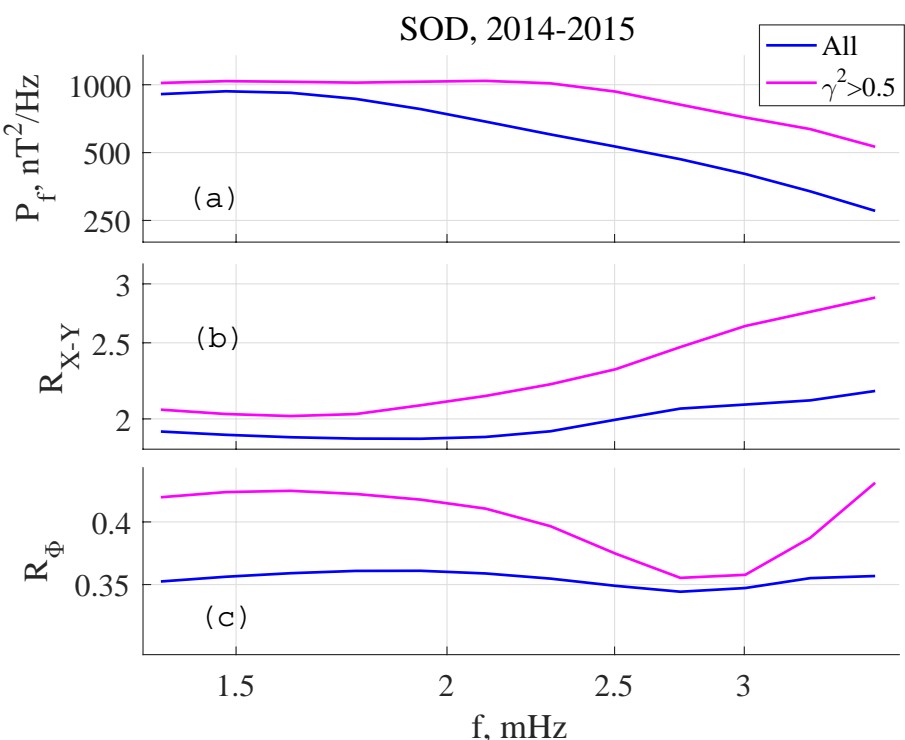

**Figure 11.** Comparison of averaged parameters of coherent with foF2 (magenta) and background (blue) Pc5/Pi3 pulsations at SOD: (a) PSD; (b) $R_{X-Y} = b_X/b_Y$ spectral ratio; (c) $R_\Phi = b_X(\text{SOD})/b_X(\text{MAS})$ spectral ratio.



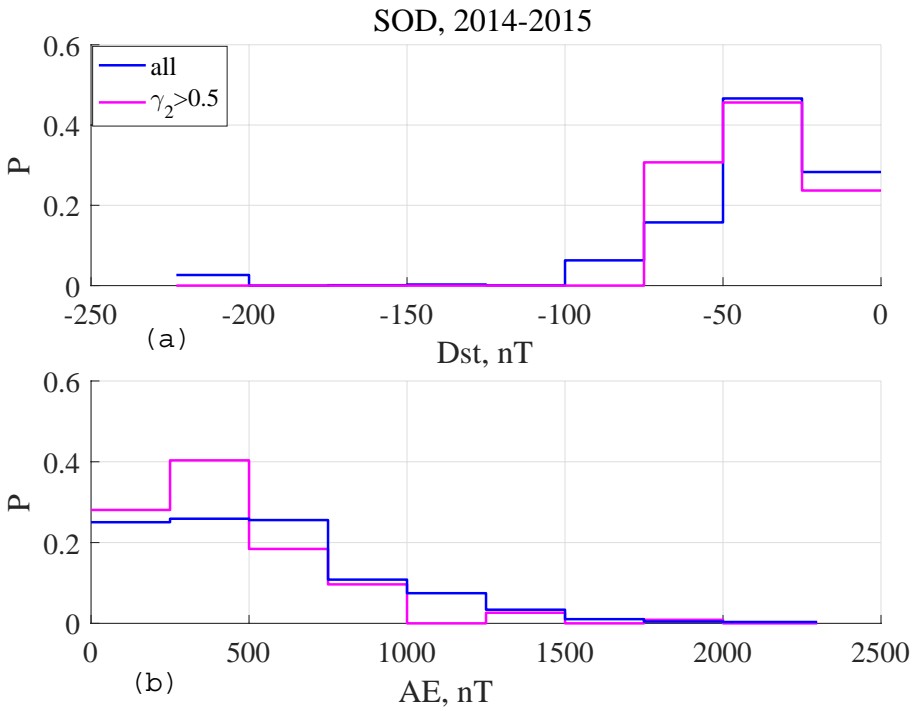

**Figure 12.** Comparison of geomagnetic indexes for coherent foF2-$b$ (magenta) and background (blue) Pc5/Pi3 pulsations at SOD: (a) distribution over Dst index; (b) distribution over AE index.





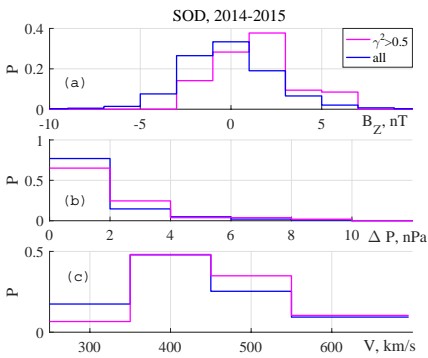

**Figure 13.** Comparison of SW/IMF parameters for coherent with foF2 (magenta) and background (blue) Pc5/Pi3 pulsations at SOD: (a) 3-hour mean IMF $B_Z$; (b) 3 hour maximal dynamic pressure variation $\Delta P$; (c) 3-hour mean SW speed $V$.

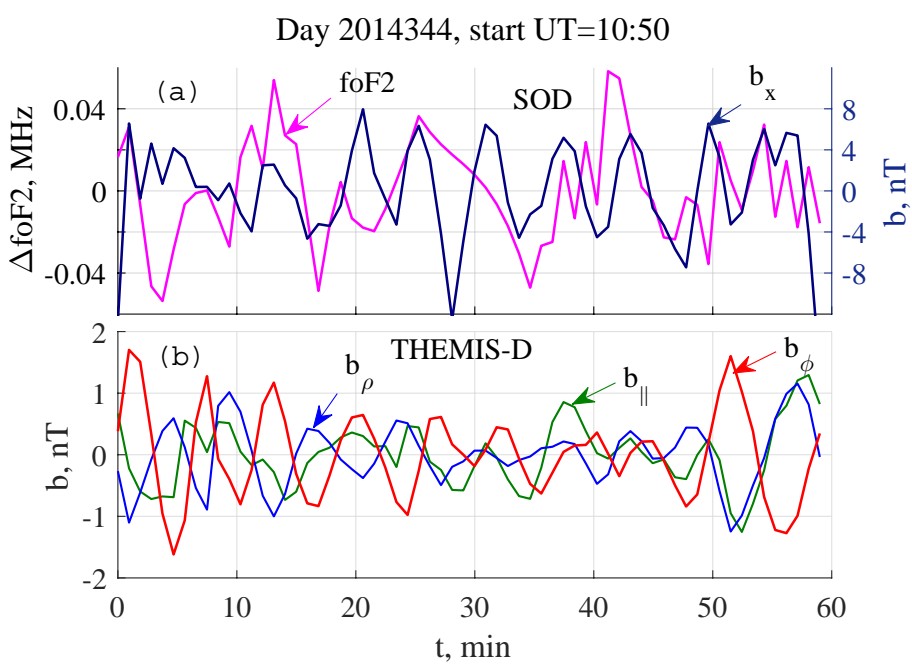

**Figure 14.** Variation of foF2 and geomagnetic pulsations at SOD (a) and THEMIS-D (b) during event 3



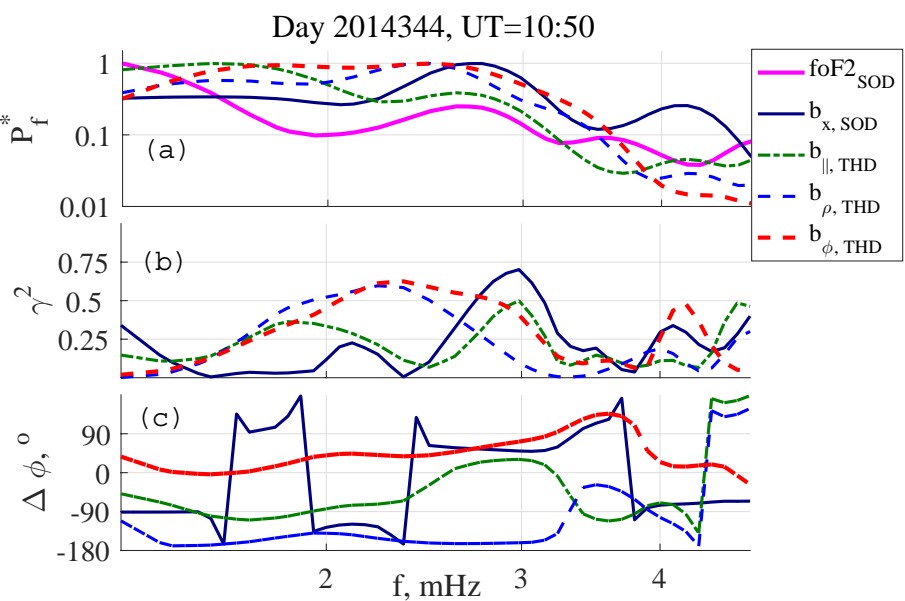

**Figure 15.** Spectral parameters for the event 3: (a) normalized PSD spectra of foF2 (solid magenta) and $b_X$ (solid blue) at SOD and 3 components at THEMIS-D (dashed lines); (b) spectral coherence between foF2 and geomagnetic pulsations at SOD and THEMIS-D; (c) phase differences for the same pairs, as at panel (b).

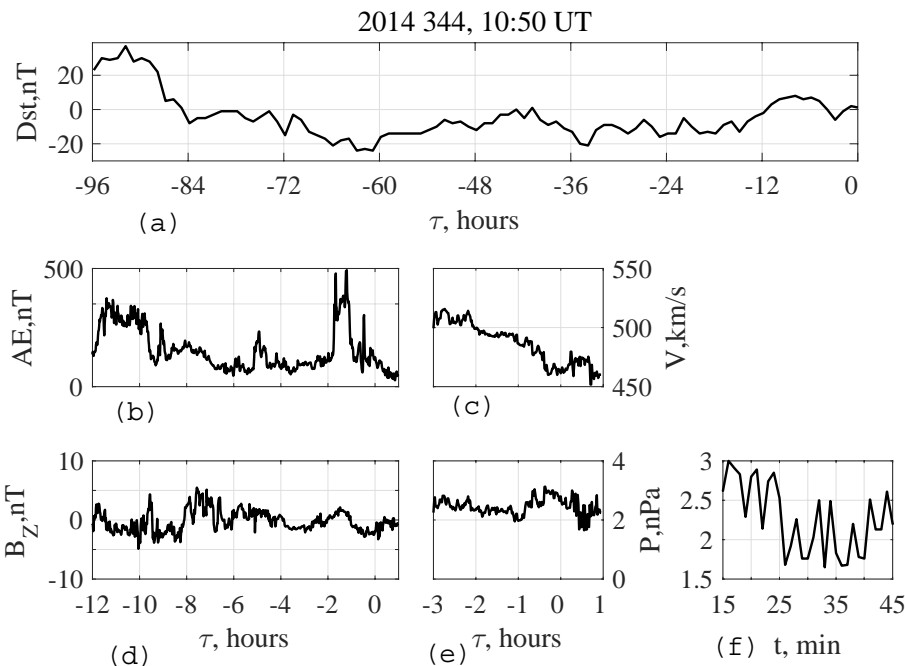

**Figure 16.** Space weather conditions for the event 3. (a) Dst index during last four days; (b) AE index during the interval and 12 hours before; (c) SW speed during the interval and 3 hours before; (d) IMF $B_Z$ during the interval and 12 hours before; (e) SW dynamic pressure during the interval and 3 hours before; (f) details of SW dynamic pressure fluctuations during the interval.

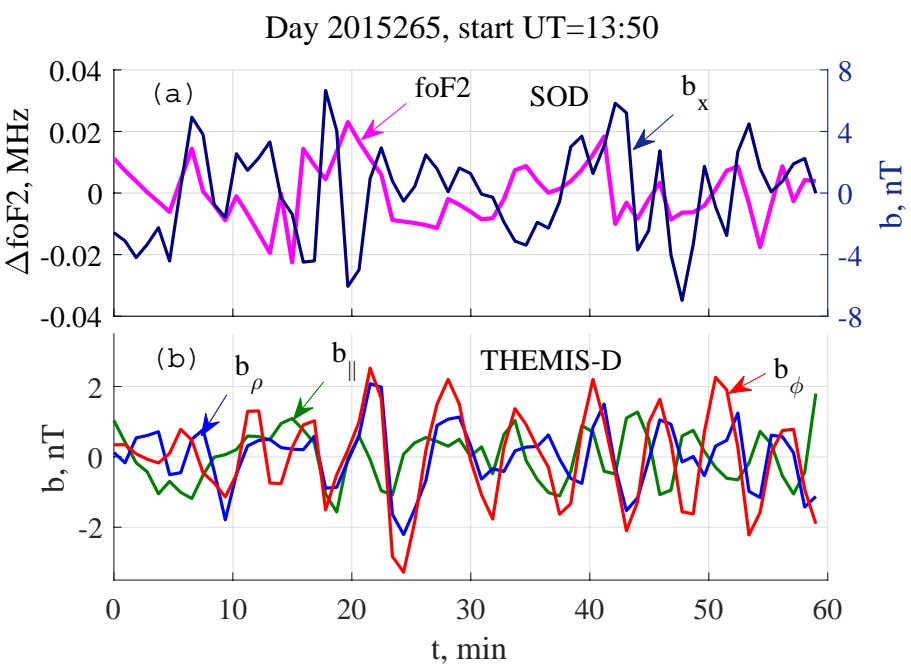

**Figure 17.** Variation of foF2 and geomagnetic pulsations at SOD (a) and THEMIS-D (b) during the event 4

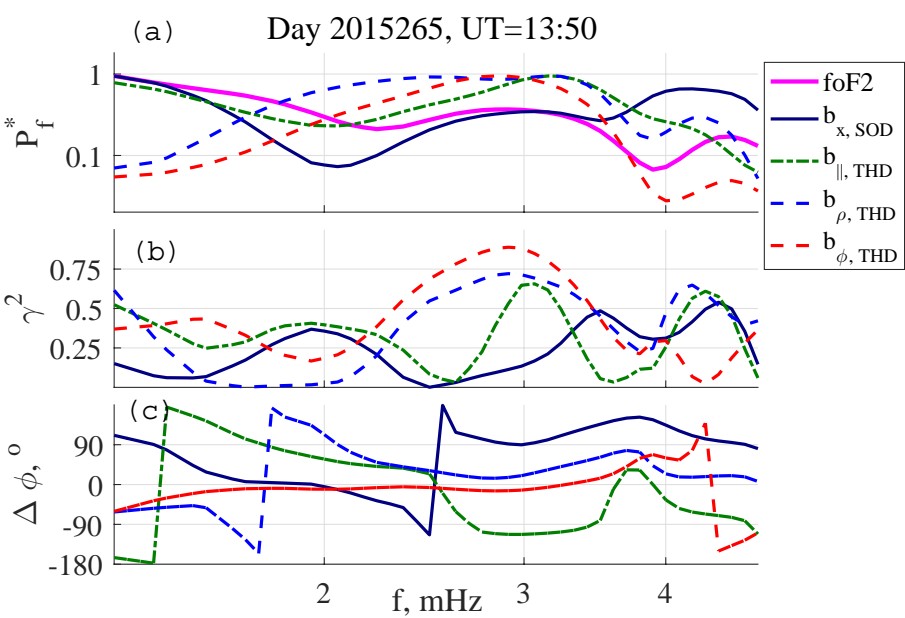

**Figure 18.** Spectral parameters for the event 4: (a) normalized PSD spectra of foF2 (solid magenta) and $b_X$ (solid blue) at SOD and 3 components at THEMIS-D (dashed lines); (b) spectral coherence between foF2 and geomagnetic pulsations at SOD and THEMIS-D; (c) phase differences for the same pairs, as at panel (b).

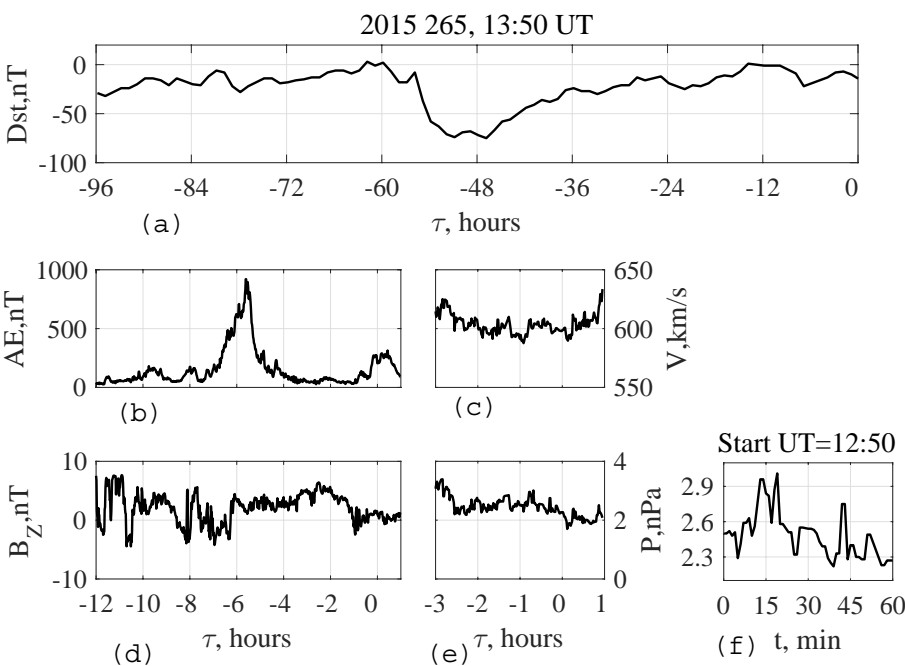

**Figure 19.** Space weather conditions for the event 3. (a) Dst index during last four days; (b) AE index during the interval and 12 hours before; (c) SW speed during the interval and 3 hours before; (d) IMF $B_Z$ during the interval and 12 hours before; (e) SW dynamic pressure during the interval and 3 hours before; (f) details of SW dynamic pressure fluctuations during the last hour before the interval.

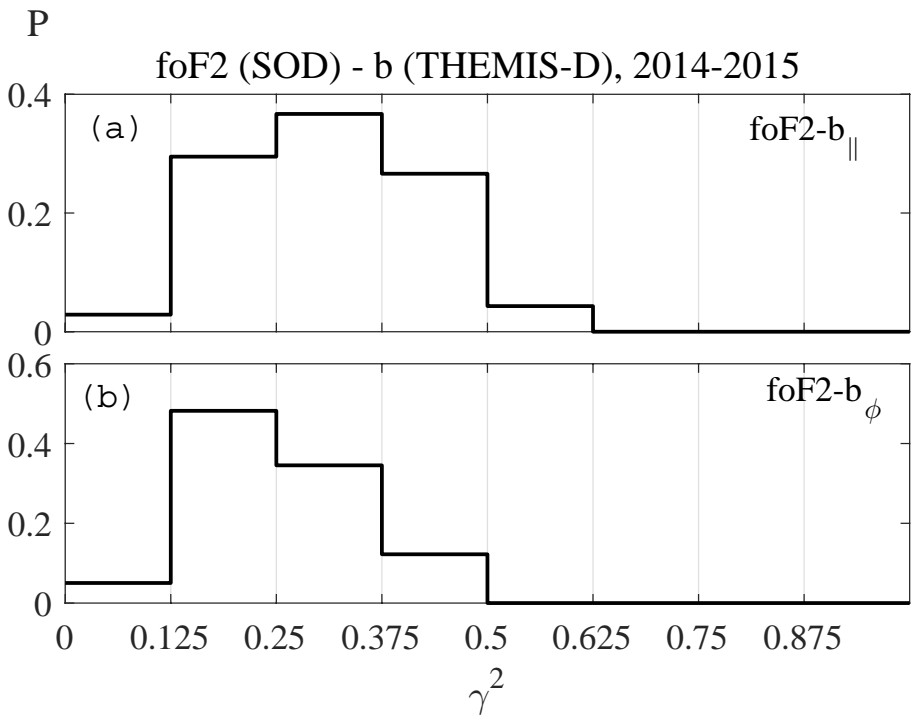

**Figure 20.** Averaged over 2 years distributions over spectral coherence between foF2 variations and geomagnetic pulsations at THEMIS-D: (a) field-aligned component $b_{\parallel}$; (b) azimuthal component $b_{\phi}$

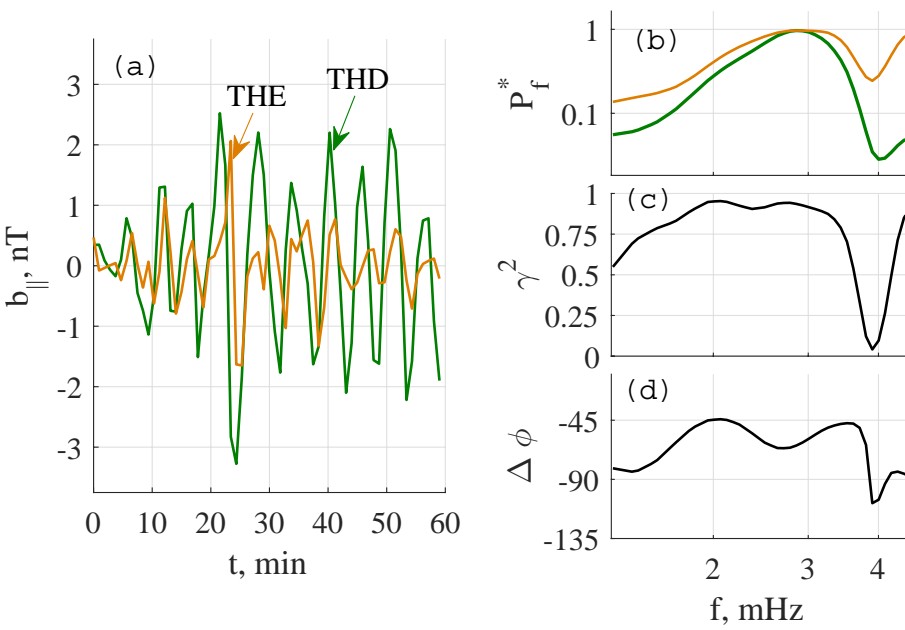

**Figure 21.** Geomagnetic pulsations at THEMISes D and E during the event 4: (a) time series for field-aligned component; (b) normalized PSD spectra; (c) spectral coherence; (d) phase difference.

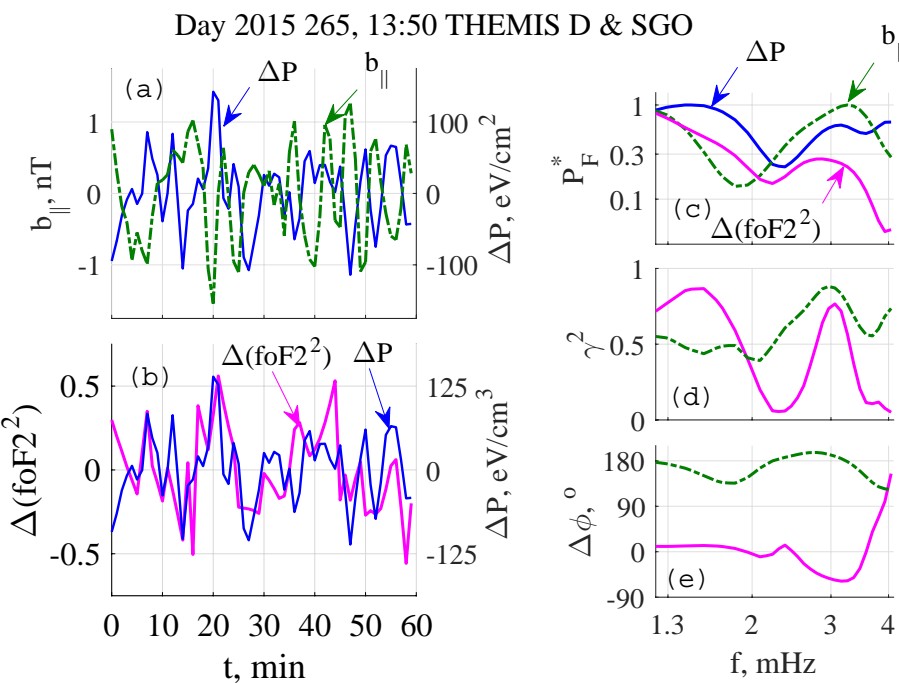

**Figure 22.** Variations of foF2$^2$ at SOD, plasma pressure $\Delta P$ and magnetic field at THEMIS-D for the event 4: (a) time series for plasma pressure and field-aligned component at THEMIS-D; (a) time series for plasma pressure at THEMIS-D; and foF2$^2$ at SOD; (c) normalized PSD spectra; (d) spectral coherence between $\Delta P$ and $b_\parallel$ at THEMIS-D (green dash) and $\Delta P$ and foF2$^2$ (magenta); (e) phase differences for the same pairs as at panel (d).





**Table 1.** Coordinates and other parameters of IMAGE stations

| Station | Geographic | | CGM | | L | MLT |
|---------|------|------|------|-------|------|----------|
| | LAT | LON | Φ | Λ | | midnight |
| SOD | 67.37 | 26.63 | 64.2 | 106.5 | 5.37 | 21:12 |
| MAS | 69.46 | 23.70 | 66.5 | 105.5 | 6.37 | 21:18 |



**Table 2.** Coordinates and other parameters of THEMIS satellites for events 3 and 4

| Event | Sat. | Date | UT | GSE ,$R_E$ | | | FP geogr. | | FP CGM | | L | MLT |
|---|---|---|---|---|---|---|---|---|---|---|---|---|
| | | YYYY DDD | hour | X | Y | Z | LAT | LON | $\Phi$ | $\Lambda$ | | |
| 3 | D | 2014 344 | 11 | 9.2 | -0.8 | -7.7 | 72.9 | 151.4 | 65.0 | 208.9 | 7.2 | 20:55 |
| 4 | D | 2015 265 | 14 | -4.6 | -10.7 | -3.3 | 64.0 | 238.2 | 69.1 | 291.5 | 8.0 | 04:23 |
| 4 | E | 2015 265 | 14 | -4.5 | -10.8 | -2.9 | 64.3 | 238.8 | 69.5 | 292.0 | 8.4 | 04:24 |