# Peer review of "Geomagnetic pulsations in the Pc5/Pi3 frequency range and fluctuations of foF2 frequency"

_Annales Geophysicae, 2019_

## Referee Comment (RC1) · Anonymous Referee #1 · 6 Jan 2020

General Comments This paper presents the relationship between oscillations of foF2 and magnetic fields in space and on the ground. Individual cases are presented to illustrate coherent and noncoherent events and statistical analyses are conducted to gain insight into the cause of the difference between the two types. foF2 oscillations are an interesting phenomenon that merits investigation. However, the study presented in this manuscript does not provide much information on the cause of the foF2 oscillations except that they are usually non-coherent with magnetic field oscillations. I do not recommend publication of the manuscript in the present form. Specific comments are listed below.

Major Comments Line 161-204, Examples. The spacecraft and the ground station are vastly separated. Unless there are other observations at different locations in the

magnetosphere and on the ground, it is difficult to evaluate if there is any causality between the oscillations in space and on the ground.

During event 4 (1350-1450 UT, 22 September (DoY = 265) 2015, THEMIS-D and THEMIS-E were in the postmidnight sector at radial distances of 12 Re, whereas the SOD station was in the postnoon sector. According to my calculation, the magnetic local time separation of the two spacecraft was 0.25 degrees. Therefore, the pi/4 phase difference of the b// oscillations at the two spacecraft (line 222) translates to an azimuthal wave number of 180. With such a high azimuthal wave number, it is impossible to have any coherence over the vast distance between SOD and THEMIS. I find the discussion on the space-ground coherence given in this section to be highly questionable.

Minor Comments

Multiple lines. Please use the standard date format "24 October 2014", or "24 October (day 297) 2014." This makes it easy for the interested readers to look at other data for comparison.

Line 38-39. Consider revising the punctuation. "This makes routine techniques based on the estimates at a given frequency-altitude non-stable, even in cases when visual detection is possible."

Line 51. What is 4-D cross?

Line 59. Where is the APPENDIX?

Line 70. "right-handed triplet?"

Line 75. Use a different symbol for dynamic pressure. "P" is already used for "signal power."

Line 88-89. "The spectrum of foF2 variations has a maximum at a frequency f = 3.2 mHz, i.e. at a f2 frequency." This is incorrect. The maximum occurs at 3.8 mHz, well

above f2.

Line 90. I suggest "...weather conditions for event 1"

Line 94-94. What is the spacecraft that measured the solar wind parameters? Where was the spacecraft located?

Lien 112. Is the distribution normalized? The vertical axis of Figure 9 is labeled "P." Use a different symbol to avoid confusion with the "P" that is already used for other quantities.

Line 129. What are "background pulsations"? Are they all events excluding the coherent events?

Line 143. "Under highly disturbed ...the probability ... vanishes" This is misleading because the occurrence distributions shown in Figure 12 are not normalized by the occurrence distributions of Dst and AE.

Line 224 and Figures 21 and 22. The THEMIS-D b// waveform does not match between Figure 21a and 22d.

Figure 5. Place the axis label on the left of the panels (c) and (e). This comment applies to other similar figures. The label for the bottom axis of panel (f) should be "tau (Greek), min".

---

## Referee Comment (RC2) · Anonymous Referee #2 · 12 Jan 2020

The paper presents several puzzling results: - The occurrence of specific Pc5 oscillations with coherent ionospheric and geomagnetic pulsations, where the relative ionospheric response $\delta N/N$ is about order of magnitude larger than the relative amplitude of a driver $\delta B/B$. - These specific oscillations are supposedly associated with global magnetospheric oscillations driven by the solar wind quasi-periodic fluctuations. However, in presented events, the satellites were in a different sector of the magnetosphere as compared with ground station.

The paper could not find a consistent description of these observations, but it is stimulating that these possibilities have been raised. In particular, anti-phase plasma and magnetic pressure variations recorded by satellite, contradicts the assumption on global fast compressional mode responsible. The small-scale mode (high-m) hardly

can be responsible for global pulsations. To confirm the global character of oscillations, additional satellite (e.g., GOES) and ground stations should be used in further studies.

Minor comments 1) In the equation (1) dimensionless expression for altitude h is desirable.

2) In Subsection 3.2.2, a description of THEMIS positions, for which the results are shown in Figure 20, is necessary. At least, L values and separation in MLT from SOD should be provided.

3) Line 140 - the end of the subsection. The decrease of probability at high geomagnetic activity may be the artifact of the method because it contradicts to the analysis cited in the m/s. This result should be discussed in a more explicit way to avoid misunderstanding.

4) Line 98: MHZ -> MHz

5) abbreviations like FP in Table 1, and notations for variables lile fïAğ|| should be explained, when they occur in the text for the first time.

6) in the introduction, authors reviewed the papers where modulation of the ionosphere by Alfven waves were treated. For completeness, it would be helpful to mention the possibility of the ionosphere modulation by fast compressional mode [Vorontsova, E., et al., Modulation of total electron content by global Pc5 waves at low latitudes, Advances in Space Research, 57, N1, 309-319, 2016, doi: 10.1016/j.asr.2015.10.041].

---

## Author Comment (AC1) · 2 Feb 2020

Dear Referee,

THank you very much for comments and suggesions. A point-to-point answer is given in the window and a colored version is given in a separate file

Major Comments Line 161-204, Examples. The spacecraft and the ground station are vastly separated. Unless there are other observations at different locations in the magnetosphere and on the ground, it is difiňĄcult to evaluate if there is any causality between the oscillations in space and on the ground.

Firstly, we agree, that we cannot discriminate between different mechanisms and physical agents, which provide wave transport from the magnetosphere to the ionosphere.

At the present stage, using the data of magnetic field and plasma pulsation phase parameters, we can only suggest that it can be some combination of coupled Alfven and one of kinetic modes, such as drift-compressional one. Secondly, in spite of the difference in MLT, the coherence between the pulsation in the magnetosphere and ionosphere in some cases is surprisingly high, which implies causality. This may be a good subject for subsequent study. We are going to include these issues into Discussion section such as: "Counter-phase variations of magnetic field and plasma pressure during event 4 are typical for compressional mode. However, the occurrence of the event 4 at recovery phase of a geomagnetic storm, satellite position in the outer magnetosphere, and high azimuthal wave number allows to attribute this pulsation in the magnetosphere to one of kinetic modes, such as drift-mirror (Pokhotelov et al., 1985) or drift-compressional (Mager et al., 2013). Its propagation at long distances in azimuthal direction can be a result of drift population of fast particles, modulated by a wave". "The wave properties found for the event 4, as well as statistical analysis of wave parameters, show features of both pure compressional or kinetic with a pronounced compressional component modes and Alfven modes. This can be a result of coupling between different modes in a non-uniform plasma (see e.g. Klimushkin and Mager (2015) and references therein). Next steps of experimental study can be done with more dense "network" of satellites in the magnetosphere, which has become available after MMS launch in 2015. Besides, measurements of differential particle flux at GOES can give information about association of observed pulsations to one of kinetic modes."

During event 4 (13501450 UT, 22 September (DoY = 265) 2015, THEMISD and THEMISE were in the postmidnight sector at radial distances of 12 Re, whereas the SOD station was in the postnoon sector. According to my calculation, the magnetic local time separation of the two spacecraft was 0.25 degrees. Therefore, the pi/4 phase difference of the b// oscillations at the two spacecraft (line 222) translates to an azimuthal wave number of 180. With such a high azimuthal wave number, it is impossible to have any coherence over the vast distance between SOD and THEMIS. I find the

discussion on the spaceground coherence given in this section to be highly questionable.

We have a little different result for the azimuthal wave number, probably due to a different technique, used for calculations. According to https://sscweb.gsfc.nasa.gov/ results, summarized in Table 2, the difference in CGM longitude is 0.5 degrees. If we do not take into account possible influence of azimuthal difference due to L-shell difference, which occur near resonant L-shell, /4 phase difference corresponds to azimuthal wave number m=90. Nevertheless, it is really very high. However, it is not obligatory mean that the wave exists only in a very narrow sector. Le et al. (2017) report on observation of globally observed high-m waves. The discussion of this problem will be added to the Discussion section such as:

"Below, we present a more detailed analysis of wave properties for the event 4. Figure 22 presents magnetograms for field-aligned component at THEMIS-D and E satellites. Although the distance between the two satellites is only 0.5 Re, and the distance between their footprints does not exceed 0.5◦ both in latitude and in longitude, the phase difference is about $\pi/4$. If we do not take into account possible influence of azimuthal difference due to L-shell difference, which occur near resonant L-shell, $\pi/4$ phase difference corresponds to azimuthal wave number m = 90. At first glance, there is a contradiction between small azimuthal scale in the magnetosphere and high coherence between magnetic pulsations at THEMIS and foF2 fluctuations at SOD. However, high m does not obligatory correspond to a narrow sector in MLT, where the wave exists. It only means that the phase changes quickly in azimuthal direction. An example of global observations of high-m pulsations has been reported by Le et al. (2017). Their observations corresponded to pulsations at a recovery phase of the magnetic storm. The event 4 in the present study also developed at the recovery phase. The question about conditions necessary, or at least favorable, for such pulsations and about physical mechanisms, which provide wave transport, should be a subject of a special study."

Minor Comments Multiple lines. Please use the standard date format "24 October 2014", or "24 October (day 297) 2014." This makes it easy for the interested readers to look at other data for comparison.

The date format is changed everyhere in the text and figure captions

Line 38-39. Consider revising the punctuation. "This makes routine techniques based on the estimates at a given frequency altitude nonstable, even in cases when visual detection is possible."

The sentence is rewritten in the following way:

"Because of these reasons, routine techniques of automatic foF2 detection can become unstable, even when visual detection is possible".

Line 51. What is 4D cross? –

The phrase is reformulated, such as

"As four fitting factors are used, a 9-point iteration procedure is organized and a parameter ... is maximized over the "cross" in space of parameters $Kt(x0, x0 -\Delta xi, x0+\Delta xi)$, where x is a point in the space of parameters, and i is a parameter number".

The notation Pt is changed with Kt in the description of criterion in the iteration procedure.

Line 59. Where is the APPENDIX?

This sentence is deleted. Now all the days are available as supplementary files.

Line 70. "righthanded triplet?"

changed

Line 75. Use a different symbol for dynamic pressure. "P" is already used for "signal power."

The problem with P is solved. Now PSD is written directly in power spectra, P is used for plasma and solar wind dynamic pressure (the latter with sw index), and D is used for empirical probability density function, because both f and P are already used for other variables.

Line 88-89. "The spectrum of foF2 variations has a maximum at a frequency f = 3.2 mHz, i.e. at a f2 frequency." This is incorrect. The maximum occurs at 3.8 mHz, well above f2.

Thank you, the description is improved.

Line 90. I suggest ". . .weather conditions for event 1"

done

Line 9494. What is the spacecraft that measured the solar wind parameters? Where was the spacecraft located?

A description of OMNI data and a link is added to section 2:

"To analyzed space weather conditions, OMNI data including interplanetary magnetic field, solar wind speed and dynamic pressure, re-calculated to the sub-solar point of the magnetosphere (Bargatze et al., 2005), are used, and Dst and AE indexes. The data are available at http://cdaweb.gsfc.nasa.gov".

Lien 112. Is the distribution normalized? The vertical axis of Figure 9 is labeled "P." Use a different symbol to avoid confusion with the "P" that is already used for other quantities.

Now D is used instead of P for empirical probability density. Yes, all the distributions are normalized.

Line 129. What are "background pulsations"? Are they all events excluding the coherent events?

Thank you, this parapgaph is rewritten, such as

"However, the coherent foF2 and geomagnetic pulsations do exist, and a question arises about the pulsation properties and external parameters, favorable for their occurrence. To answer this question, the geomagnetic pulsations at SOD for which bX $-$foF2 coherence is high ($\gamma2 > 0.5$) are compared with all the intervals, selected for spectral analysis of foF2 fluctuations at SOD during 21 months from April of 2014 till the end of 2015 (see complimentary files for full information). To avoid the influence of different seasonal and diurnal variations of the selected pulsations and average pulsation properties, the statistics for all the pulsations is calculated with the weight functions calculated from the seasonal and diurnal variations of coherent pulsations. Figure 11 illustrates the difference between coherent and pulsations and averaged properties of all pulsations for three parameters: PSDbx (Figure 11a), PSD ratio RXY = PSDbx/PSDby (Figure 11b), and the bX PSD ratio along a magnetic meridian $R\Phi$ = PSDbx($\Phi$)/PSDbx($\Phi+\Delta\Phi$) (Figure 11c). The latter is calculated for SOD-MAS station pair (MAS station is located nearly at the same magnetic meridian, but it is shifted in 2ảŮę northward). PSDbx for coherent pulsations is enriched with frequenciesf >2 mHz in comparison with the background pulsations. In this frequency band, RXY also increases and $R\Phi$ demonstrates a non-monotonous dependence on frequency with minimum at f = 2.7 mHz and growth at f $\geq$ 3 mHz. These features are only weakly seen in averaged $R\Phi$(f) dependence for all pulsation intervals. To understand, what space weather conditions are favorable for generation of coherent bX $-$foF2 pulsations, we compare the geomagnetic indexes and SW/IMF conditions for intervals when coherent bX $-$foF2 were registered with all the intervals analyzed. The influence of seasonal and diurnal variation was eliminated in the same manner, as for pulsation parameters. We use for the analysis the 4-day minimum Dst and 6-hour maximal AE, as Pc5 amplitudes are maximal at recovery phase of geomagnetic storms (Posch et al., 2003), and auroral substorms are followed by Pi3 pulsations (Kleimenova et al., 2002) and Pc5 waves with high azimuthal and intermediate wavenumbers (Zolotukhina et al., 2008; Mager et al., 2019). The results for Dst and AE indexes are summarized in Figure 12. Coherent pulsations tend to occur under moderate geomagnetic and auroral activity. The most favorable Dst interval is from $-100$ to $-50$ nT (Figure 12a), and for AE index it is from 250 to 500 nT (Figure 12b)."

Although, because of rare occurrence of coherent pulsations, there is almost no difference between coherent and background, defined as "all – coherent", coherent intervals are compared with all the selected intervals (i.e. those, for which quality of foF2 detection was enough for spectral estimates), and now it is written in the text in a more explicit way.

Line 143. "Under highly disturbed . . .the probability . . . vanishes" This is misleading because the occurrence distributions shown in Figure 12 are not normalized by the occurrence distributions of Dst and AE.

Thank you, this point is changed. The other problem is the selection procedure which also leads to decrease of weight of disturbed intervals and this is also mentioned in the text in the following way

"Actually, the selection procedure, used in the present study to detect intervals with clearly seen foF2 fluctuations, is limited by quiet and moderately disturbed geomagnetic conditions. This leads to low probabilities to detect foF2 fluctuations at Pc5/Pi3 frequencies under highly disturbed conditions. This result naturally follows from the condition of existence of clear layer structure, necessary for the pulsation detection procedure. During geomagnetic storms detection of the foF2 variations is often impossible because of enhanced ionization in the lower ionospheric layers (E and/or D)".

Line 224 and Figures 21 and 22. The THEMISD b// waveform does not match between Figure 21a and 22d.

Thank you, it was really an other interval, by mistake shown in Figure 21. All the intervals are checked and improved.

Figure 5. Place the axis label on the left of the panels (c) and (e). This comment

applies to other similar figures. The label for the bottom axis of panel (f) should be "tau (Greek), min".

done

Please also note the supplement to this comment:
https://www.ann-geophys-discuss.net/angeo-2019-155/angeo-2019-155-AC1-supplement.pdf

---

## Author Comment (AC2) · 2 Feb 2020

Dear referee, thank you very much for helpful comments. A point-by-point answer is given in the window and a colored version is put as a supplementary file.

The paper presents several puzzling results: The occurrence of specifidc Pc5 oscillations with coherent ionospheric and geomagnetic pulsations, where the relative ionospheric response $\Delta N/N$ is about order of magnitude larger than the relative amplitude of a driver $\Delta B/B$. These specific oscillations are supposedly associated with global magnetospheric oscillations driven by the solar wind quasiperiodic fluctuations. However, in presented events, the satellites were in a different sector of the magnetosphere as compared with ground station. The paper could not find a con-

sistent description of these observations, but it is stimulating that these possibilities have been raised. In particular, antiphase plasma and magnetic pressure variations recorded by satellite, contradicts the assumption on global fast compressional mode responsible. The smallscale mode (highm) hardly can be responsible for global pulsations. To confirm the global character of oscillations, additional satellite (e.g., GOES) and ground stations should be used in further studies.

Actually, high azimuthal wave number does not contradict to a possibility of global scale of oscillations. A far, as we know now direct observations of global high-m pulsations have been reported only for storm-time pulsations (Le et al., 2017). We plan to study pulsations found in the present study to discriminate between properties of quiet-time pulsations and those developed at storm-time using a more dense "network" of satellite observations, which became available after MMS launch in 2015. Besides, measurements of differential electron flux at GOES can give information about association of observed pulsations to one of kinetic modes, such as drift-compressional. However, these problems require separate studies. In the present MS, we have extended the Discussion section and where these points are now briefly discussed, e.g.

"At first glance, there is a contradiction between small azimuthal scale in the magnetosphere and high coherence between magnetic pulsations at THEMIS and foF2 fluctuations at SOD. However, high m does not obligatory correspond to a narrow sector in MLT, where the wave exists. It only means that the phase changes quickly in azimuthal direction. An example of global observations of high-m pulsations has been reported by Le et al. (2017). Their observations corresponded to pulsations at a recovery phase of the magnetic storm. The event 4 in the present study also developed at the recovery phase. The question about conditions necessary, or at least favorable, for such pulsations and about physical mechanisms, which provide wave transport, should be a subject of a special study." ... "The wave properties found for the event 4, as well as statistical analysis of wave parameters, show features of both pure compressional or kinetic with a pronounced compressional component modes and Alfven modes. This

can be a result of coupling between different modes in a non-uniform plasma (see e.g. Klimushkin and Mager (2015) and references therein). Next steps of experimental study can be done with more dense "network" of satellites in the magnetosphere, which has become available after MMS launch in 2015. Besides, measurements of differential particle flux at GOES can give information about association of observed pulsations to one of kinetic modes."

A picture with coherent variations of THEMIS-D magnetic pulsation and electron flux measured at GOES-13 separated by 4.5 hours from THEMIS – D during the event 4 is given below as an illustration.

Minor comments

1) In the equation (1) dimensionless expression for altitude h is desirable. – done

2) In Subsection 3.2.2, a description of THEMIS positions, for which the results are shown in Figure 20, is necessary. At least, L values and separation in MLT from SOD should be provided.

A picture of footprint distribution for THEMIS-D in CGM coordinates is included (Figure 20) and a description of relative position of THEMIS footprints and SOD is added to the text.

3) Line 140 the end of the subsection. The decrease of probability at high geomagnetic activity may be the artifact of the method because it contradicts to the analysis cited in the m/s. This result should be discussed in a more explicit way to avoid misunderstanding.

Now a desctiption is given in the following redaction:

"Actually, the selection procedure, used in the present study to detect intervals with clearly seen foF2 fluctuations, is limited by quiet and moderately disturbed geomagnetic conditions. This leads to low probabilities to detect foF2 fluctuations at Pc5/Pi3 frequencies under highly disturbed conditions. This result naturally follows from the

condition of existence of clear layer structure, necessary for the pulsation detection procedure. During geomagnetic storms detection of the foF2 variations is often impossible because of enhanced ionization in the lower ionospheric layers (E and/or D)".

4) Line 98: MHZ > MHz

done

5) abbreviations like FP in Table 1, and notations for variables should be explained, when they occur in the text for the first time.

done

6) in the introduction, authors reviewed the papers where modulation of the ionosphere by Alfven waves were treated. For completeness, it would be helpful to mention the possibility of the ionosphere modulation by fast compressional mode [Vorontsova, E., et al., Modulation of total electron content by global Pc5 waves at low latitudes, Advances in Space Research, 57, N1, 309319, 2016, doi: 10.1016/j.asr.2015.10.041].

The citation is added to Introduction

"Observations of pulsations in the total electron content (TEC) are rather rare (Pilipenko et al., 2014a, b; Watson et al., 2015; Vorontsova, 2016)" . . . "An effect of TEC modulation by ULF wave at low latitudes reported by Vorontsova (2016) is important because it is observed far away from the resonant L-shells and zones where kinetic modes can occur due to wave-particle interaction. This allows to identify observed pulsations as fast magnetosonic mode".

References

Le, G., et al.: Global observations of magnetospheric high-m poloidal waves during the 22 June 2015 magnetic storm, Geophys. Res. Lett., 44, 3456–3464, doi:10.1002/2017GL073048, 2017.

[Figure]

Please also note the supplement to this comment:
https://www.ann-geophys-discuss.net/angeo-2019-155/angeo-2019-155-AC2-
supplement.pdf
* * *
[Figure]

**Fig. 1.**